# Connecting Multi-modal Contrastive Representations

**Zehan Wang**[1]    **Yang Zhao**[2]   **Xize Cheng**[1]   **Haifeng Huang**[1]   **Jiageng Liu**[1]   **Li Tang**[1]
**Linjun Li**[1]   **Yongqi Wang**[1]   **Aoxiong Yin**[1]   **Ziang Zhang**[1]   **Zhou Zhao**[1, 3*]

[1]Zhejiang University   [2]ByteDance   [3]Shanghai AI Laboratory
{wangzehan01}@zju.edu.cn

## Abstract

Multi-modal Contrastive Representation (MCR) learning aims to encode different modalities into a semantically aligned shared space. This paradigm shows remarkable generalization ability on numerous downstream tasks across various modalities. However, the reliance on massive high-quality data pairs limits its further development on more modalities. This paper proposes a novel training-efficient method for learning MCR without paired data called Connecting Multi-modal Contrastive Representations (C-MCR). Specifically, given two existing MCRs pre-trained on $(\mathcal{A}, \mathcal{B})$ and $(\mathcal{B}, \mathcal{C})$ modality pairs, we project them to a new space and use the data from the overlapping modality $\mathcal{B}$ to aligning the two MCRs in the new space. Meanwhile, since the modality pairs $(\mathcal{A}, \mathcal{B})$ and $(\mathcal{B}, \mathcal{C})$ are already aligned within each MCR, the connection learned by overlapping modality can also be transferred to non-overlapping modality pair $(\mathcal{A}, \mathcal{C})$. To unleash the potential of C-MCR, we further introduce a semantic-enhanced inter- and intra-MCR connection method. We first enhance the semantic consistency and completion of embeddings across different modalities for more robust alignment. Then we utilize the inter-MCR alignment to establish the connection, and employ the intra-MCR alignment to better maintain the connection for inputs from non-overlapping modalities. To demonstrate the effectiveness of C-MCR, we take the field of audio-visual and 3D-language learning as examples. Specifically, we connect CLIP and CLAP via texts to derive audio-visual representations, and integrate CLIP and ULIP via images for 3D-language representations. Remarkably, without using any paired data, C-MCR for audio-visual achieves state-of-the-art performance on audio-image retrieval, audio-visual source localization, and counterfactual audio-image recognition tasks. Furthermore, C-MCR for 3D-language also attains advanced zero-shot 3D point cloud classification accuracy on ModelNet40. Our project page is available at https://c-mcr.github.io/C-MCR/

## 1   Introduction

Multi-modal Contrastive Representation (MCR) learning aims to map inputs from different modalities to a shared representation space. With the impressive generalization performance of vision-language contrastive pre-training models [1, 2, 3, 4] demonstrated on various downstream tasks [5, 6, 7, 8, 9, 10], learning MCR spaces between multiple modalities has become a promising area of research, attracting increasing attention [11, 12, 13, 14, 15].

However, the generalization ability of MCR primarily benefits from the accessibility of massive data pairs from the web. For modalities where obtaining semantically matching data pairs is significantly more costly, the representations directly learned from limited data pairs are unreliable. On the other

---

*Corresponding author.

37th Conference on Neural Information Processing Systems (NeurIPS 2023).

hand, these modality pairs with little direct paired data often have a large number of paired data with the same intermediate modality. For example, although audio-visual data are often vague, paired data of audio-language and language-image are sufficient and semantically explicit. Similarly, while 3D point-language pairs are rare, 3D point-image and image-language data are extensive.

Consider that there are already many MCRs between modalities with sufficient paired data. In this paper, we propose Connecting Multi-modal Contrastive Representations (C-MCR), a novel training-efficient and paired-data-free MCR learning method that extends the learned alignment knowledge in existing MCRs to more modalities. With regard to the overlapping modality, its representations in two MCRs are just different data views sharing the same inherent semantics. So we can take them as positive pairs to connect different MCRs. As modalities within each MCR are semantically aligned, the connections built from overlapping modalities can also be applied to non-overlapping modalities. The advantages of our C-MCR are two-fold: **(1) Flexible.** C-MCR enables MCR learning on modalities with limited paired data. More importantly, C-MCR treats each learned MCR space as a node and the overlapping modalities between different MCRs as links. Connecting the various isolated MCRs greatly extends the obtained multi-modal alignment knowledge, and discovers generalized contrastive representations of broader modalities. **(2) Training-Efficient.** Since C-MCR simply reprojects the learned representations into a new space, only two simple projectors are learnable during training. The training parameters and costs for connecting existing MCRs are very small.

However, two factors impede the acquisition of a robust and transferable connection: Firstly, embeddings in MCR spaces are incapable of comprehensively reflecting all the semantic information of the input, and this loss of meaning would be inherited and amplified, thereby compromising the robustness of the connection. Secondly, as discussed in [16], MCR spaces exhibit a *modality gap* phenomenon, i.e., the embeddings of different modalities are located in two completely separate regions in each MCR space. This poses a challenge for maintaining the connection based on overlapping modality while facing inputs from non-overlapping modalities.

Considering the above challenges, we propose a semantic-enhanced inter- and intra-MCR connection method. During training, the copious amounts of easily accessible unpaired unimodal data are first encoded into embeddings in two MCR spaces. We inject Gaussian noise into all the embeddings to mitigate the semantic bias, enhance the semantic completeness, and improve robustness. For directly quantifying the modality gap and the relationship between non-overlapping modalities, we exploit the inherent multi-modal alignment in MCR spaces to cluster semantic consistent embeddings and bridge different modalities. With the above strategies, we align the semantic-enhanced embeddings across different MCR spaces in a contrastive manner to establish the connection. To preserve the connection for inputs from non-overlap modalities, we realign the semantic similar embeddings across modalities within each MCR space to alleviate the modality gap.

Our main contributions are summarized as follows:

(1) We propose Connecting Multi-modal Contrastive Representations (C-MCR), a novel paired-data-free and training-efficient method for MCR learning. By connecting existing MCR spaces with simple projectors, we can mine the multi-modal alignment knowledge in existing MCR space, and extend MCRs on more modalities that lack large-scale high-quality data pairs.

(2) We further propose a semantic-enhanced inter- and intra-MCR connection method to unleash our C-MCR. This approach establishes a transferable connection between two MCR spaces via overlapping modality and maintains it for non-overlapping modalities.

(3) To demonstrate the effectiveness of C-MCR, we connect the CLIP and CLAP through texts to acquire audio-visual representations, and interage CLIP and 3D-image MCR space (ULIP) via images for 3D-language representations. Remarkably, without requiring any pair data or fine-tuning, C-MCR for audio-visual achieves state-of-the-art performance on six datasets across three downstream audio-visual tasks. Furthermore, C-MCR for 3D-language also attains advanced zero-shot 3D point cloud classification accuracy on ModelNet40.

## 2 Related Work

**Multi-modal Contrastive Representation learning.** Multi-modal contrastive representation focuses on learning separate unimodal encoders for different modalities, which can map inputs from different modalities into a shared representation space. These models are pre-trained on large-scale paired data

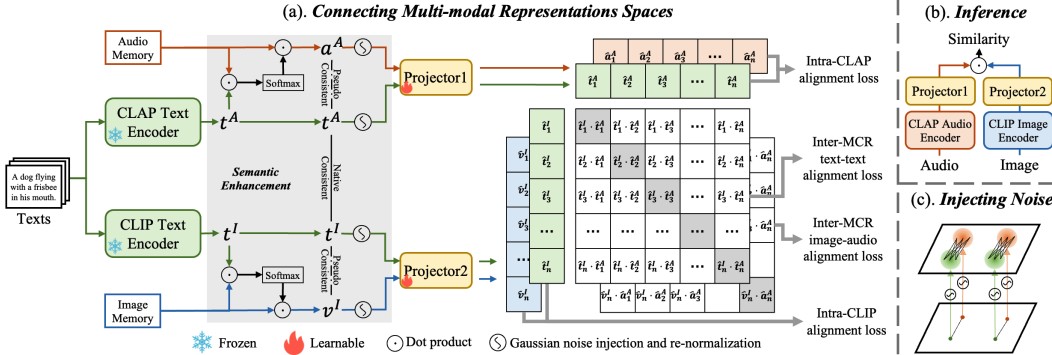

Figure 1: **The pipeline of connecting CLIP and CLAP using our C-MCR.** During training, we take text as input and encode it with frozen CLAP and CLIP text encoders, respectively. Audio(image) memory is generated by encoding lots of unimodal audio(image) data by pre-trained audio(image) encoder. Semantic enhancement enriches the semantic consistency and completion of embeddings. Then two projectors map embeddings to a new shared representation space where inter- and intra-MCR alignment establishes and maintains a stable connection between CLAP and CLIP. During inference, the audio and image are inputted to the corresponding encoder and projector.

using a contrastive loss. Recent vision-language contrastive pre-training models, such as CLIP [1] and ALIGN [2], demonstrate impressive zero-shot retrieval and classification performance and remarkable generalization capability across diverse downstream tasks [5, 6, 7, 8, 9, 10]. Inspired by the success of vision-language models, contrastive representation learning across more modalities has garnered increasing attention. CLAP [12, 11] construct a contrastive language-audio pre-training model by collecting large-scale audio-text pairs from diverse data sources. ULIP [17, 18] collect and generate 3D-image-text triplet data via 3D rendering and image captioning, and learn an extra 3D encoder for existing vision-language space. AudioCLIP [19] and WAV2CLIP [20] leverage the pre-trained CLIP image encoder and acquire audio-visual representations by training on audio-image pairs from AudioSet [21] and VGGSound [22], respectively. For certain modality pairs, such as audio-visual and 3D-language, the pre-training model's generalization capability is restricted by ambiguous or limited paired data. Our proposed method introduces a novel method for better contrastive representation learning on these modalities.

**Audio-Visual Learning.** Audio-visual learning [23] aims to exploit the relationship between audio and visual modalities, which is an essential part of intelligent multi-modal perception research. Previous methods primarily focus on learning specific audio-visual downstream tasks (such as retrieval [24, 25, 26], localization [27, 28, 29, 30, 31, 32, 33, 34, 35, 36], or generation [37, 38, 39, 40, 41, 42, 43, 44]) within limited domains, based on the manually cleaned small-scale datasets. Recently, several large-scale audio-image datasets [21, 22] collected from the web are proposed. However, these datasets contain many noisy image-audio pairs due to the ambiguous nature of both images and audio and the presence of non-visible sounds and silent objects in videos. Consequently, the generalization ability of audio-visual contrastive representation [19, 20] learned from these datasets is limited. Our C-MCR reduces the need for larger-scale high-quality data pairs. By extending the knowledge in CLIP and CLAP models, we acquire powerful audio-visual contrastive representations that exhibit powerful generalization capabilities across various downstream tasks.

**3D-language Learning.** 3D vision is an important way for robots to perceive the rich semantic and spatial information of the real world. The 3D-language learning including recognition [45, 46, 47, 48], localization [49, 50, 51, 52, 53], question-answer [54, 55] and general conversation [56, 57], have attracted increasing attentions. 3D-language contrastive representations are vital for the further development of 3D-language learning. However, due to the scarcity of 3D-language paired data, the development of 3D-language representation is limited. Recent ULIP [17, 18] focus on generating 3D-image-text triplet data, but they are still limited by the relatively low quality of training datasets. C-MCR gets rid of the dependence on 3D-language pairing data, and instead connects the reliable 3D-visual representation of ULIP and the visual-language representation of CLIP via images to obtain a more robust 3D-language contrastive representation.

# 3 Approach

In this section, we take connecting CLIP and CLAP for audio-visual, as an example to introduce C-MCR. As depicted in Figure 1 (a), we utilize two projectors to connect CLIP and CLAP through texts. Before delving into our method, we first introduce the mathematical formulations and revisit the multi-modal contrastive learning in Section 3.1. Then we discuss our semantic enhancement approach for robust representation alignment in Section 3.2. This is followed by the inter-MCR alignment to establish the connection between CLIP and CLAP in Section 3.3, and the intra-MCR alignment to ensure the connection can be maintained for image-audio inputs in Section 3.4.

## 3.1 Background

**Problem formulation.** For text inputs, the embeddings obtained by CLIP and CLAP encoder can be denoted as $\mathbf{t}^I \in \mathbb{R}^c$ and $\mathbf{t}^A \in \mathbb{R}^d$ respectively. Our C-MCR method aims to leverage the inherent consistency between $\mathbf{t}^I$ and $\mathbf{t}^A$ and the multi-modal alignment in MCR to learn two projectors $f_1(\cdot)$ and $f_2(\cdot)$ that map the representations from CLIP and CLAP to a new shared representation space. The connection between CLIP and CLAP, learned from texts, can effectively accommodate audio and image inputs.

**Multi-modal Contrastive Learning.** Given $N$ paired instances from two different modalities, we map the $i$-th pair to L2-normalized embeddings $\mathbf{x}_i$ and $\mathbf{z}_i$ via two encoders. Multi-modal contrastive learning aims to maximize the cosine similarity between $\mathbf{x}_i$ and $\mathbf{z}_i$ and minimize the cosine similarity between $\mathbf{x}_i$ and $\mathbf{z}_j$ where $i \neq j$. The contrastive loss can be formulated as:

$$L_{cons} = -\frac{1}{2}\frac{1}{N}\sum_{i=1}^{N}\left[\log\frac{\exp(\mathrm{sim}(\mathbf{x}_i,\mathbf{z}_i)/\tau)}{\sum_{j=1}^{N}\exp(\mathrm{sim}(\mathbf{x}_i,\mathbf{z}_j)/\tau)} + \log\frac{\exp(\mathrm{sim}(\mathbf{z}_i,\mathbf{x}_i)/\tau)}{\sum_{j=1}^{N}\exp(\mathrm{sim}(\mathbf{z}_i,\mathbf{x}_j)/\tau)}\right] \quad (1)$$

where $\tau$ is the temperature parameter and the $\mathrm{sim}(\cdot,\cdot)$ is the operator for cosine distance. The contrastive loss is based on multi-modal data pairs, and the generalization of the learned representation relies on the scale and quality of the data pairs. To extend contrastive representation learning on more modalities, we propose using overlapping modalities to connect two learned MCR spaces and extend the learned multi-modal alignment knowledge to non-overlapping modalities.

## 3.2 Semantic Enhancement

To achieve more robust and comprehensive alignment, we enhance the semantics from two perspectives: inter-modality semantic consistency and intra-modality semantic completion.

**Inter-modality Semantic Consistency.** CLIP and CLAP have already learned shared image-text and audio-text representations. To better quantify the modality gap in the MCR space and directly explore the correlation between audio and image, we first utilize the inherent modality alignment properties of CLIP and CLAP to generate semantically consistent embeddings across modalities. Specifically, we encode massive unpaired images and audio using the CLIP image encoder and CLAP audio encoder, respectively. All the obtained image embeddings are served as image memory $\mathbf{V} = \{\mathbf{v}_1, \mathbf{v}_2, ..., \mathbf{v}_N\}$ and the audio embeddings are audio memory $\mathbf{A} = \{\mathbf{a}_1, \mathbf{a}_2, ..., \mathbf{a}_M\}$, where $N, M$ indicate the number of images and audios. Considering $i$-th text embeddings $\mathbf{t}_i^I$ and $\mathbf{t}_i^A$, we can generate image embedding $\mathbf{v}_i^I$ and audio embedding $\mathbf{a}_i^A$ that are semantically similar to $i$-th text.

$$\mathbf{v}_i^I = \sum_{k=1}^{N}\frac{\exp(\mathrm{sim}(\mathbf{t}_i^I,\mathbf{v}_k)/\tau_1)}{\sum_{j=1}^{N}\exp(\mathrm{sim}(\mathbf{t}_i^I,\mathbf{v}_j)/\tau_1)} * \mathbf{v}_k; \quad \mathbf{a}_i^A = \sum_{k=1}^{M}\frac{\exp(\mathrm{sim}(\mathbf{t}_i^A,\mathbf{a}_k)/\tau_1)}{\sum_{j=1}^{M}\exp(\mathrm{sim}(\mathbf{t}_i^A,\mathbf{a}_j)/\tau_1)} * \mathbf{a}_k \quad (2)$$

The $\tau_1$ is the temperature hyperparameter. By dynamically absorbing information from memories based on semantic similarity to the text embeddings $\mathbf{t}_i^I$ and $\mathbf{t}_i^A$, we can generate more diverse and accurate semantically-consistent embeddings $\mathbf{v}_i^I$ and $\mathbf{a}_i^A$.

**Intra-modality Semantic Completion.** The semantics in the original input data are often complex, and some information is inevitably lost when encoding it into the MCR space. When connecting and aligning existing representation spaces, this loss and bias of meaning will be inherited and amplified, affecting the robustness of alignment. To enhance the semantic completeness of each embedding,

we propose to serve Gaussian noise as an information augmentation method. Specifically, we add zero-mean Gaussian noises into the embeddings and re-normalize them to the unit hypersphere:

$$\tilde{\mathbf{t}}^I = \text{Normalize}(\mathbf{t}^I + \boldsymbol{\theta}_1); \qquad \tilde{\mathbf{v}}^I = \text{Normalize}(\mathbf{v}^I + \boldsymbol{\theta}_2)$$
$$\tilde{\mathbf{t}}^A = \text{Normalize}(\mathbf{t}^A + \boldsymbol{\theta}_3); \qquad \tilde{\mathbf{a}}^A = \text{Normalize}(\mathbf{a}^A + \boldsymbol{\theta}_4) \tag{3}$$

where noise items $\boldsymbol{\theta}_1, \boldsymbol{\theta}_2 \in \mathbb{R}^c$ and $\boldsymbol{\theta}_3, \boldsymbol{\theta}_4 \in \mathbb{R}^d$ are sampled from zero-mean gaussian distribution with variance $\sigma^2$, and they are not learnable.

Since the MCRs are L2 normalized, all embeddings are distributed on a unit sphere. As illustrated in Figure 1 (c), each embedding can be viewed as a point on the unit sphere's surface. The addition of Gaussian noise can transform the point into a small sphere, and re-normalizing projects the small sphere onto a circle on the surface of the unit sphere. Hence, aligning two embeddings with noise forces the model to acquire the ability to align all the embeddings within the two circles. In the MCR space, the closer two embeddings are to each other, the more similar their semantics are. Embeddings within the same circle share similar general semantics, and the semantics represented by the circle are more comprehensive and robust than the original embedding.

### 3.3 Inter-MCR Alignment

To establish the connection between two MCRs, we project the semantic-enhanced embeddings from CLIP and CLAP space to a new shared space via two learnable projectors $f_1(\cdot)$ and $f_2(\cdot)$, respectively.

$$\hat{\mathbf{t}}^I = f_1(\tilde{\mathbf{t}}^I); \ \ \hat{\mathbf{v}}^I = f_1(\tilde{\mathbf{v}}^I); \ \ \hat{\mathbf{t}}^A = f_2(\tilde{\mathbf{t}}^A); \ \ \hat{\mathbf{a}}^A = f_2(\tilde{\mathbf{a}}^A) \tag{4}$$

In the newly projected space, our objective is to ensure that embeddings with similar semantics from different MCR spaces are in close proximity to each other. The $(\mathbf{t}_i^I, \mathbf{t}_i^A)$ from the same text is naturally semantic consistent, and it can be considered as a ground-truth pair label. Besides, there is pseudo consistency in $(\mathbf{v}_i^I, \mathbf{t}_i^I)$ and $(\mathbf{a}_i^A, \mathbf{t}_i^A)$ due to the multi-modal alignment properties in CLIP and CLAP. Thus the $(\hat{\mathbf{v}}^I, \hat{\mathbf{a}}^A)$ derived from $(\mathbf{t}_i^I, \mathbf{t}_i^A)$ can be viewed as a pseudo pair label. For a robust and stable connection of the two MCR, we propose to align both $(\hat{\mathbf{t}}^I, \hat{\mathbf{t}}^A)$ and $(\hat{\mathbf{v}}^I, \hat{\mathbf{a}}^A)$. The text-text contrastive loss $L_{ttc}$ and audio-visual contrastive loss $L_{avc}$ are defined as:

$$L_{ttc} = -\frac{1}{2}\frac{1}{B}\sum_{i=1}^{B}\left[\log\frac{\exp(\text{sim}(\hat{\mathbf{t}}_i^I, \hat{\mathbf{t}}_i^A)/\tau_2)}{\sum_{j=1}^{B}\exp(\text{sim}(\hat{\mathbf{t}}_i^I, \hat{\mathbf{t}}_j^A)/\tau_2)} + \log\frac{\exp(\text{sim}(\hat{\mathbf{t}}_i^A, \hat{\mathbf{t}}_i^I)/\tau_2)}{\sum_{j=1}^{B}\exp(\text{sim}(\hat{\mathbf{t}}_i^A, \hat{\mathbf{t}}_j^I)/\tau_2)}\right] \tag{5}$$

$$L_{avc} = -\frac{1}{2}\frac{1}{B}\sum_{i=1}^{B}\left[\log\frac{\exp(\text{sim}(\hat{\mathbf{v}}_i^I, \hat{\mathbf{a}}_i^A)/\tau_3)}{\sum_{j=1}^{B}\exp(\text{sim}(\hat{\mathbf{v}}_i^I, \hat{\mathbf{a}}_j^A)/\tau_3)} + \log\frac{\exp(\text{sim}(\hat{\mathbf{a}}_i^A, \hat{\mathbf{v}}_i^I)/\tau_3)}{\sum_{j=1}^{B}\exp(\text{sim}(\hat{\mathbf{a}}_i^A, \hat{\mathbf{v}}_j^I)/\tau_3)}\right] \tag{6}$$

$B$ corresponds to the batch, and $\tau_2$, $\tau_3$ are the temperature hyperparameters. The inter-MCR alignment loss $L_{inter}$ is the combination of the two contrastive losses:

$$L_{inter} = L_{ttc} + L_{avc} \tag{7}$$

The $L_{ttc}$ and $L_{avc}$ are complementary to each other. The semantics between $(\mathbf{t}^I, \mathbf{t}^A)$ are highly consistent, thus the connection learned from them is much more robust, but their alignment is indirect for audio-visual representation. On the other hand, $(\mathbf{v}^I, \mathbf{a}^A)$ pairs are directly beneficial to audio-visual representation learning, but their semantic coherence is less reliable. Note that since the semantic consistency in $(\mathbf{v}^I, \mathbf{a}^A)$ is derived from $(\mathbf{t}^I, \mathbf{t}^A)$, the connection learned from pseudo pair $(\mathbf{v}^I, \mathbf{a}^A)$ can still be considered as being established via overlapping modalities.

### 3.4 Intra-MCR Alignment

As discussed in [16], there exists a phenomenon known as the *modality gap* in MCR spaces. Although the embeddings from different modalities are semantically aligned in an MCR space, they are distributed in entirely distinct regions of the representation space. This implies that the more stable connection learned from $(\mathbf{t}_i^I, \mathbf{t}_i^A)$ may not accommodate the inputs from audio and image.

To better maintain the connection, we propose closing the modality gap and guaranteeing that embeddings from different modalities with similar semantics are distributed in the same region of the representation space. The analysis in [16] suggests that the repulsive structure in contrastive loss

preserves the modality gap. Inspired by this observation, we derive the intra-MCR alignment loss by removing the repulsive structure in the contrastive loss. As introduced in 3.1, a typical contrastive item can be formulated as:

$$-\log \frac{\exp(\text{sim}(\mathbf{x}_i, \mathbf{z}_i)/\tau)}{\sum_{j=1}^{N} \exp(\text{sim}(\mathbf{x}_i, \mathbf{z}_j)/\tau)} = \underbrace{-\text{sim}(\mathbf{x}_i, \mathbf{z}_i)/\tau}_{pull\ positive\ close} + \underbrace{\log \sum_{j=1}^{N} \exp(\text{sim}(\mathbf{x}_i, \mathbf{z}_j)/\tau)}_{push\ negative\ away} \tag{8}$$

We only retain the mechanism of pulling samples closer together and remove the repulsive effect between negative pairs, which helps to close the modality gap in the newly learned MCR space. In the L2-normalized MCR space, there are $(\mathbf{x}_i - \mathbf{y}_i)^T (\mathbf{x}_i - \mathbf{y}_i) = 2(1 - \mathbf{x}_i^T \mathbf{y}_i)$. After removing the gradient-irrelevant constant terms, our intra-MCR alignment loss $L_{intra}$ can be expressed as:

$$L_{intra} = \frac{1}{2} \frac{1}{B} \sum_{i=1}^{B} (\|\hat{\mathbf{t}}_i^I - \hat{\mathbf{v}}_i^I\|_2 + \|\hat{\mathbf{t}}_i^A - \hat{\mathbf{a}}_i^A\|_2) \tag{9}$$

By realigning text-guided cross-modal semantically consistent embeddings in each MCR space, i.e., aligning $(\hat{\mathbf{t}}_i^I, \hat{\mathbf{v}}_i^I)$ for CLIP and $(\hat{\mathbf{t}}_i^A, \hat{\mathbf{a}}_i^A)$ for CLAP, the modality gap between embeddings from same MCR can be effectively alleviated in the new space. As a result, the more stable connection provided by Equation 5 can be maintained for audio-visual inputs.

### 3.5 Training and Inference

During training, all pre-trained encoders in CLIP and CLAP are frozen to preserve the semantic correspondences between image-text and audio-text, and only the two projectors are learnable. To make the training more efficient, we pre-extract the text embeddings $\mathbf{t}_i^I$ and $\mathbf{t}_i^A$. Since the semantic enhancements are training-free, the inter-modality semantic consistency strategy can also be pre-computed, and the semantically consistent image $\mathbf{v}_i^I$ and audio embedding $\mathbf{a}_i^A$ are stored offline.

We apply a combination of inter- and intra-MCR alignment loss to optimize the two projectors for establishing a stable connection between CLIP and CLAP representation spaces, formulated as:

$$L = L_{inter} + \lambda L_{intra} \tag{10}$$

$\lambda$ is the hyper-parameter to balance the two terms.

During inference, as shown in Figure 1 (b), the image embedding in CLIP and the audio embedding in CLAP can be mapped into a shared space through corresponding projectors. The cosine scores in this space reflect the semantic similarity between images and audio.

## 4 Experiments

### 4.1 Details of Connecting CLAP and CLIP

**Text Datasets.** We collected texts from three sources: image-text datasets (COCO [58] and CC3M [59]), video-text datasets (MSRVTT [60], MAD [61]), and audio-text datasets (AudioCap [62], Clotho [63]), to ensure that the texts contain sufficient visual, action, and audio information. To avoid overfitting visual information, we randomly selected one million descriptions from CC3M. In summary, the texts from image-text, video-text, and audio-text are 1.66M, 0.58M, and 77K, respectively, and there are 2.33M texts in total.

**Audio/Image Memory.** AudioSet [21] provides a vast collection of audio snippets from YouTube videos. All 1.8M audio data in the training set are encoded by the CLAP audio encoder to serve as the audio memory. ImageNet1K [64] is a large-scale image recognition dataset. We encoded all the 1.3M images in the train set of ImageNet1K using the CLIP image encoder to construct the image memory. It is worth noting that no annotations related to the audio and images are used.

**Implementation Details.** We employ a frozen pre-trained CLIP ViT-B/32 model [1] and CLAP model [13]. We adopt simple multi-layer perceptrons as our projectors $f_1(\cdot)$ and $f_2(\cdot)$. The $\tau_1$, $\tau_2$ and $\tau_3$ in Euqation 2, 5 and 6 are all set to 1/100. The variance $\sigma^2$ of the noises in Equation 3 is set as 0.004. The hyper-parameter $\lambda$ in Equation 10 is set to 0.1. We train our projectors for 36 epochs using a batch size of 10240. We use the AdamW optimizer with the initial learning rate $1e-3$ and the cosine learning rate decay strategy.

## 4.2  Details of Connecting ULIP and CLIP

**Image Datasets.** The image dataset used for connecting ULIP and CLIP is ImageNet1K [64], total 1.3M images without any annotations.

**Text/3D Memory.** We use the same 2.33M text dataset as described in Section 4.1, to construct the corresponding text memory. The 3D object point clouds from the training set of Objaverse [48] are utilized to construct 3D memory, 0.8M samples in total.

**Implementation Details.** We employ a frozen pre-trained CLIP ViT-B/32 model [1], and ULIP-2 PointBERT model [65, 13] pre-trained on ULIP-Objaverse triplets. The structure of the projector and the temperature parameters remain the same in Section 4.1. The variance $\sigma^2$ of the noises in Equation 3 is set as 0.002. The hyper-parameter $\lambda$ in Equation 10 is set to 0.4. We train our projectors for 24 epochs using a batch size of 8192. We also use the AdamW optimizer with the initial learning rate $5e-3$ and the cosine learning rate decay strategy.

## 4.3  Evaluation of Audio-Visual Representations

### 4.3.1  Downstream Audio-Visual Tasks

We assess the quality of audio-visual representations on three downstream audio-visual tasks in a zero-shot manner. More details about the datasets and implementation details are in Appendix.

**Audio-Image Retrieval.** It contains two subtasks: image-to-audio retrieval (I2A) and audio-to-image retrieval (A2I). We assess the zero-shot image-audio retrieval on the AVE [66] and Flickr-SoundNet [32]. Due to the small size of the test sets in both datasets, we utilized all available data in the train, eval, and test sets for evaluation, resulting in 4095 samples for AVE and 5000 samples for Flickr-SoundNet. For zero-shot inference, we encode all audio and images into our newly learned audio-visual MCR space and computed the cosine similarity for all audio-image pairs. The mAP, Top-1, and Top-5 metrics are used to evaluate retrieval accuracy.

**Audio-Visual Source Localization.** Audio-visual source localization aims to localize the visual sound sources in an image. The test sets of widely-used VGGSS [67] and MUSIC [68] benchmarks are employed for evaluation. To enable zero-shot inference, we first use a pre-trained object detector [3] to extract object proposals from the images and calculate the cosine similarity between each proposal and audio in our representations space. The proposal with the highest similarity score is token as the final prediction. We adopt Consensus Intersection over Union (cIoU) and Area Under Curve(AUC) metrics following [28, 69].

**Counterfactual Audio-Image Recognition.** For the non-visible sounds and images with silent objects, this task requires a model to distinguish the semantically unpaired audio-image from audio-image pairs. During the zero-shot inference phase, we employ an object detector [3] to extract object proposals from the image. Subsequently, for each image, the proposal with the highest matching score is considered the predicted object, and the matching score is regarded as the confidence score for this prediction. Experiments are conducted on the Extended VGGSS (Ex-VGGSS) [69] and Extended Flickr-SoundNet (Ex-FlickrNet) [69], and the comparison is based on the Average Precision (AP) and maximum F1 (Max-F1) metrics following [69].

In summary, these three tasks can evaluate a group of audio-visual contrastive representations from various perspectives. Audio-image retrieval is employed to assess the ability to match coarse-grained images and audio, audio-visual source localization is used to evaluate the ability to match fine-grained objects and audio, and counterfactual audio-image recognition is used to evaluate the understanding and reasoning ability of audio and visual inputs.

### 4.3.2  Analysis on Zero-shot Image-Audio Retrieval

We compared our model with AudioCLIP [19] and WAV2CLIP [20] which are contrastively pre-trained on image-audio pairs from AudioSet [21] and VGGSound [22], respectively. Results in Table 1 demonstrate that C-MCR achieves state-of-the-art zero-shot retrieval performance. Besides, the generalization ability of AudioCLIP and WAV2CLIP is not stable. For instance, WAV2CLIP performs well on AVE but poorly on Flickr-SoundNet, while AudioCLIP achieves good results on Flickr-SoundNet but poor accuracy on AVE. Similar situations can also be observed in Table 2. In contrast, our C-MCR exhibits stronger and more stable generalization ability. Moreover, since

Table 1: Zero-shot audio-image retrieval results on AVE and Flickr-SoundNet. A-V pairs stands for whether training on paired audio-visual data; Tr. Param for Trainable parameters number.

| Method | A-V Pairs | Tr. Param | AVE | | | | | | Flickr-SoundNet | | | | | |
| | | | A2I | | | I2A | | | A2I | | | I2A | | |
| | | | mAP | R@1 | R@5 | mAP | R@1 | R@5 | mAP | R@1 | R@5 | mAP | R@1 | R@5 |
|---|---|---|---|---|---|---|---|---|---|---|---|---|---|---|
| Random | - | - | 0.25 | 0.02 | 0.12 | 0.25 | 0.02 | 0.12 | 0.17 | 0.02 | 0.06 | 0.17 | 0.02 | 0.06 |
| WAV2CLIP | ✓ | 11.7M | 2.80 | 0.76 | 3.08 | 4.01 | 1.14 | 4.42 | 2.52 | 0.58 | 3.16 | 3.47 | 1.12 | 4.34 |
| AudioCLIP | ✓ | 134.1M | 0.98 | 0.22 | 0.85 | 2.50 | 1.00 | 2.83 | 3.10 | 1.00 | 4.02 | 4.43 | 1.58 | 5.92 |
| C-MCR | ✗ | 2.1M | **4.11** | **1.25** | **4.54** | **4.13** | **1.25** | **4.44** | **4.57** | **1.38** | **5.40** | **4.92** | **1.58** | **5.98** |

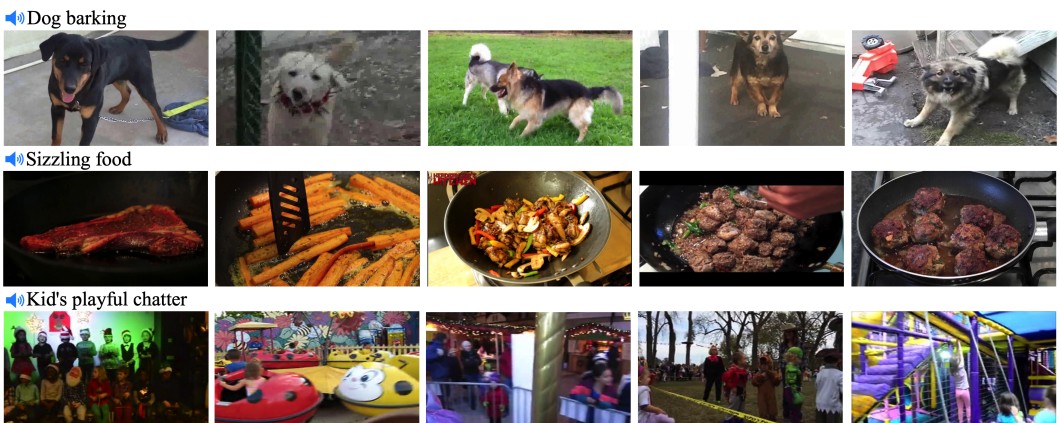

🔊 Dog barking

🔊 Sizzling food

🔊 Kid's playful chatter

Figure 2: Visualization of audio-to-image retrieval on AVE and Flickr-SoundNet.

C-MCR does not utilize any paired data and has much fewer learnable parameters, AudioCLIP and WAV2CLIP are not truly "fair" baselines to compare C-MCR with. Nevertheless, C-MCR still demonstrates superior performance compared to these pre-trained audio-visual models. Figure 2 provides a few visualizations of audio-to-image retrieval.

### 4.3.3 Analysis on Zero-shot Audio-Visual Source Localization

Table 2 presents the zero-shot audio-visual source localization performance on MUSIC-Solo and VGGSS datasets and the comparison with previous audio-visual source localization methods. Remarkably, despite not using any audio-visual paired data and any fine-tuning, C-MCR demonstrates state-of-the-art performance, achieving a relative improvement of around 25% over the previous leading methods. Additionally, to demonstrate that the improvements are not by introducing the powerful object detector, we also performed the same zero-shot inference using audio-visual representations in AudioCLIP and WAV2CLIP. These methods exhibit unstable generalization performance on the two datasets, and our C-MCR demonstrates a significantly better overall performance than both. These results show that the state-of-the-art performances in audio-visual source localization mainly benefit from the stronger and more robust fine-grained audio-visual matching capability.

### 4.3.4 Analysis on Zero-shot Counterfactual Audio-Image Recognition

Table 3 shows the comparisons on counterfactual audio-image recognition. Our C-MCR significantly outperforms previous methods that trained on the training set and exhibits overall improvement compared to other audio-visual representation models. The state-of-the-art performance on zero-shot counterfactual audio-image recognition further demonstrates the superiority of our method in understanding the deep semantic relationship between audio and visual modalities.

Table 2: Zero-shot audio-visual source localization on MUSIC-Solo and VGGSS.

| Method | MUSIC-Solo cIoU | AUC | VGGSS cIoU | AUC |
|---|---|---|---|---|
| Attention [32] | 37.20 | 38.70 | 17.10 | 28.70 |
| DMC [31] | 29.10 | 38.00 | 23.90 | - |
| DSOL [33] | 51.40 | 43.60 | 29.91 | - |
| TURN [28] | 33.70 | 45.20 | 34.60 | 39.10 |
| EZ-VSL [36] | - | - | 38.85 | 39.54 |
| SLAVC [69] | - | - | 39.80 | - |
| WAV2CLIP [20] | 47.49 | 53.80 | 36.91 | 39.58 |
| AudioCLIP [19] | 30.56 | 37.16 | 43.93 | 45.96 |
| C-MCR(Ours) | **53.78** | **56.09** | **48.08** | **48.69** |

Table 3: Zero-shot counterfactual audio-image recognition on Ex-VGGSS and Ex-FlickrNet.

| Method | Ex-VGGSS AP | Max-F1 | Ex-FlickrNet AP | Max-F1 |
|---|---|---|---|---|
| Attention [32] | 6.70 | 13.10 | 15.98 | 24.00 |
| DMC [31] | 11.53 | 20.30 | 25.56 | 41.80 |
| DSOL [33] | 16.84 | 25.60 | 38.32 | 49.40 |
| OGL [36] | 18.73 | 30.90 | 40.20 | 55.70 |
| EZ-VSL [36] | 27.71 | 34.60 | 48.75 | 56.80 |
| SLAVC [69] | 34.46 | 41.50 | 52.15 | 60.10 |
| WAV2CLIP [20] | 33.86 | 47.69 | 60.54 | 66.20 |
| AudioCLIP [19] | 42.59 | 55.43 | 72.78 | 71.98 |
| C-MCR(Ours) | **50.91** | **58.98** | **73.67** | **74.02** |

## 4.4 Evaluation of 3D-language Representations

In order to verify the performance of the 3D-language representation obtained by connecting ULIP-2 and CLIP, we evaluate the zero-shot 3D point cloud classification accuracy on ModelNet40, and the results are shown in Table 4. Our C-MCR achieves state-of-the-art zero-shot classification results compared with methods trained on 3D-language data. The advanced performance in the 3D-language field further demonstrates the great potential of C-MCR to learn contrastive representations for modalities lacking paired data.

Table 4: Zero-shot 3D point cloud classification results on ModelNet40.

| Method | Top1 | Top3 | Top5 |
|---|---|---|---|
| ReCon [70] | 61.2 | 73.9 | 78.1 |
| CG3D [15] | 48.7 | 60.7 | 66.5 |
| ULIP [17] | 60.4 | 79.0 | 84.4 |
| ULIP-2 [18] | **74.0** | 86.5 | 90.0 |
| C-MCR | 64.9 | **87.0** | **92.8** |

## 4.5 Ablation Studies

We conduct ablation studies on audio-image retrieval over AVE and Flickr-SoundNet to examine the effectiveness of our method. All the results are presented in Table 5 and Figure 3.

**Semantic Consistency.** We use a softmax function to softly aggregate embeddings in memory and produce the semantic consistent embedding in Equation 2. For comparison, Row I selects the embedding in the memory with the highest similarity as generated embedding, while Row J randomly selects an embedding in the memory. Compared to Row I, the significantly better results in Rows J and K highlight the necessity of inter-modality semantic consistency. Compared to the approach of hardly selecting one embedding, as in Row I, our soft clustering of memories slightly improves the performance by generating more diverse embeddings.

**Semantic Completion.** By comparing H and K, we can find that semantic bias in the MCR space indeed dramatically affects the learning of connections, and adding noise to embeddings can effectively alleviate this issue by enhancing semantic completeness and robustness. The results in G and K demonstrate that our re-normalized operator in Equation 3 is beneficial for learning alignment on the unit sphere. Moreover, in Figure 3, we vary the variance $\sigma^2$ of noises and report its effect. Generally, the performance is not sensitive to changes in $\sigma^2$.

**Inter-MCR alignment.** Comparison between D, E, and K demonstrates that the connection learned from the native text-text pairs is much more crucial than from the pseudo-consistent audio and visual embeddings, and using both native text-text pairs and pseudo audio-visual pairs produces the best performance. Furthermore, if no connections are made, as in Row F, the image and audio embeddings would have no semantic relationship since the original CLIP and CLAP spaces are isolated.

**Intra-MCR alignment.** Results in A, B, and K indicate that alignment within either CLIP or CLAP can provide a relatively reliable connection, and aligning both leads to even better results. Conversely, not aligning CLIP and CLAP as in C results in a connection not well adapted to audio-visual input. More importantly, the results in C are similar to those in Row D, where connections are not learned from text-text pairs. This observation indicates that the primary function of intra-MCR alignment is to alleviate the modality gap, thereby enabling the more stable connections learned from text-text pairs to adapt to audio-visual inputs.

Table 5: Ablation studies in AVE and Flickr-SoundNet retrieval. We report the "mAP" metric on both A2I and I2A subtasks. Re-norm stands for the re-normalized operator in Equation 3; CLIP for intra-CLIP alignment item in Equation 9; CLAP for intra-CLAP alignment item in Equation 9; FlickrNet for Flickr-SoundNet dataset.

| Rows | Consistency | Completion | | Inter-MCR | | Intra-MCR | | AVE | | FlickrNet | |
| | | Re-norm | Noise | $L_{ttc}$ | $L_{avc}$ | CLAP | CLIP | A2I | I2A | A2I | I2A |
|---|---|---|---|---|---|---|---|---|---|---|---|
| A | softmax | ✓ | ✓ | ✓ | ✓ | ✗ | ✓ | 4.09 | 4.11 | 4.52 | 4.71 |
| B | softmax | ✓ | ✓ | ✓ | ✓ | ✓ | ✗ | 3.97 | 4.08 | 4.44 | 4.79 |
| C | softmax | ✓ | ✓ | ✓ | ✓ | ✗ | ✗ | 3.14 | 3.22 | 3.63 | 3.51 |
| D | softmax | ✓ | ✓ | ✗ | ✓ | ✓ | ✓ | 3.28 | 3.30 | 3.69 | 3.50 |
| E | softmax | ✓ | ✓ | ✓ | ✗ | ✓ | ✓ | 4.09 | 4.10 | 4.42 | 4.54 |
| F | softmax | ✓ | ✓ | ✗ | ✗ | ✓ | ✓ | 0.22 | 0.23 | 0.18 | 0.19 |
| G | softmax | ✗ | ✓ | ✓ | ✓ | ✓ | ✓ | 3.70 | 3.88 | 4.57 | 4.62 |
| H | softmax | ✗ | ✗ | ✓ | ✓ | ✓ | ✓ | 2.77 | 2.37 | 2.72 | 2.57 |
| I | argmax | ✓ | ✓ | ✓ | ✓ | ✓ | ✓ | 4.01 | 3.99 | 4.49 | 4.52 |
| J | ✗ | ✓ | ✓ | ✓ | ✓ | ✓ | ✓ | 2.62 | 2.84 | 2.52 | 2.76 |
| K | softmax | ✓ | ✓ | ✓ | ✓ | ✓ | ✓ | **4.11** | **4.13** | **4.57** | **4.92** |

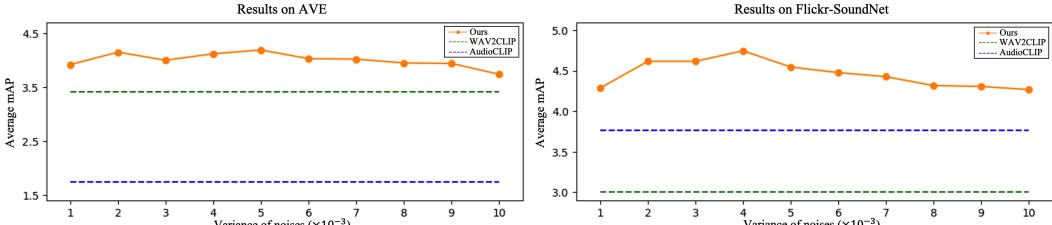

Figure 3: Effect of different variance $\sigma^2$ of the noises in Equation 3 on AVE and Flickr-SoundNet retrieval. The average mAP is the mean value of the mAP in I2A and A2I subtasks.

## 5 Conclusion

This paper proposes Connecting Multi-modal Contrastive Representation (C-MCR), a new flexible and training-efficient method for learning multi-modal contrastive representation. C-MCR eliminates the need for large-scale, high-quality data pairs and instead extends the acquired multi-modal alignment knowledge in existing MCRs. By connecting existing MCRs via overlapping modality, we are able to discover more generalized contrastive representations across a broader range of modalities. Experimentally, we learn state-of-the-art audio-visual contrastive representations by connecting CLIP and CLAP through texts, and advanced 3D-language representations by connecting CLIP and ULIP via images. Despite not utilizing any paired data, the representations obtained by C-MCR significantly outperform previous representations learned from data pairs on different downstream tasks.

## Acknowledgments

This work was supported in part by National Key R&D Program of China under Grant No.2022ZD0162000, National Natural Science Foundation of China under Grant No.62222211, No.62072397 and No.61836002.

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

# A Ablation Study about Text Dataset.

We conduct more experiments on audio-image retrieval with training texts from different sources. Furthermore, we provide insights about selecting training data when employing our C-MCR to connect other MCRs. As discussed in Sec 4.1, our training texts are collected from three sources: image-text datasets (COCO [58] and CC3M [59]), video-text datasets (MSRVTT [60] and MAD [61]), and audio-text datasets (AudioCap [62] and Clotho [63]).

In Table 4, we exclude the text data from image-text, video-text, and audio-text datasets, respectively. The results demonstrate that combining data from all three sources achieves the best performance. Furthermore, our findings suggest that the information in video-text datasets is relatively less important than in image-text and audio-text datasets. When applying our C-MCR to connect other MCRs, it is critical to collect overlapping modality data associated with information from non-overlapping modalities to

Figure 4: Ablation studies about text datasets for training.

| Dataset | AVE | | Flickr | |
|---|---|---|---|---|
| | A2I | I2A | A2I | I2A |
| Full | **4.11** | **4.13** | **4.57** | **4.92** |
| w/o image-text | 3.83 | 3.76 | 3.97 | 3.91 |
| w/o video-text | 4.04 | 4.05 | 4.41 | 4.78 |
| w/o audio-text | 4.07 | 3.88 | 4.31 | 4.59 |

ensure robust connections. The data from the pre-training datasets used by MCR could serve as an appropriate starting point. Combining the overlapping modality data from these sources ensures that the data used for constructing connections contains sufficient information from non-overlapping modalities. Additionally, this data is easily accessible and scalable, which greatly enhances the practicality of our C-MCR.

# B Downstream Task Details.

## B.1 Audio-Image Retrieval.

We consider two datasets for this task: AVE [66] and Flickr-SoundNet [32], both of which consist of semantically matched image and audio pairs that were manually curated. To more comprehensively and stably reflect the retrieval capability of the model, we use all available data in these two datasets for evaluation, resulting in 4,095 samples for AVE and 5,000 samples for Flickr-SoundNet.

## B.2 Audio-Visual Source Localization.

We conduct experiments on the VGGSS [67] and MUSIC [68] datasets. VGGSS is derived from VGGSound, and its test set comprises 5,158 audio-image pairs. MUSIC consists of 489 untrimmed videos of musical solos spanning 11 instrument categories for testing. It is worth noting that we use the category names from the COCO dataset as prompts to enable the open-vocabulary object detector GLIP [3] to extract object proposals.

## B.3 Counterfactual Audio-Image Recognition.

The Extended Flickr-SoundNet [69] and Extended VGGSS [69] are constructed by adding 250 and 5,158 negative samples to the test sets of the original Flickr-SoundNet and VGGSS datasets, respectively. The prompts used for the object detector GLIP [3] are also the category names from the COCO dataset. We evaluate the counterfactual Audio-Image Recognition performance using the Maximum F1 (Max-F1) and Average Precision (AP) metrics, following [69]. During inference, for the $i$-th image-audio pair, the proposal with the highest matching score with the audio is considered the predicted object, and its matching score is considered the confidence score $c_i$. The CIoU of the predicted object is denoted as $IoU_i$. The ground-truth map is denoted as $\mathcal{G}_i$, and the ground-truth maps of negative samples are $\emptyset$. Under these definitions, the true positives $\mathcal{TP}$, false positives $\mathcal{FP}$, and false negatives $\mathcal{FN}$ are computed as:

$$
\begin{aligned}
\mathcal{TP}(\gamma, \delta) &= \{i | \mathcal{G}_i \neq \emptyset, IoU_i > \gamma, c_i > \delta\} \\
\mathcal{FP}(\gamma, \delta) &= \{i | \mathcal{G}_i \neq \emptyset, IoU_i \leq \gamma, c_i > \delta\} \cup \{i | \mathcal{G}_i = \emptyset, c_i > \delta\} \\
\mathcal{FN}(\gamma, \delta) &= \{i | \mathcal{G}_i \neq \emptyset, c_i \leq \delta\}
\end{aligned}
\tag{11}
$$

where $\gamma$ is the threshold of $IoU$ and $\delta$ is the threshold of confidence score. Following previous work, the $\gamma$ is set as 0.5. The F1 score can be represented as:

$$F1(\gamma, \delta) = \frac{2 * \text{Precision}(\gamma, \delta) * \text{Recall}(\gamma, \delta)}{\text{Precision}(\gamma, \delta) + \text{Recall}(\gamma, \delta)} \tag{12}$$

where

$$\text{Precision}(\gamma, \delta) = \frac{|\mathcal{TP}(\gamma, \delta)|}{|\mathcal{TP}(\gamma, \delta)| + |\mathcal{FP}(\gamma, \delta)|}; \ \ \text{Recall}(\gamma, \delta) = \frac{|\mathcal{TP}(\gamma, \delta)|}{|\mathcal{TP}(\gamma, \delta)| + |\mathcal{FN}(\gamma, \delta)|} \tag{13}$$

In accordance with [69], we calculate F1 scores for all values of $\delta$ and report the maximum F1 score (Max-F1). Average Precision (AP) is another commonly used metric in object detection, its computation is detailed in [58, 69].

## C  Model Configurations.

Table 6: Model configurations of projectors.

| Module | Block | $C_{in}$ | $C_{out}$ |
|--------|-------|----------|-----------|
| Projector1 | Linear | 512 | 1024 |
|  | BatchNorm1D | 1024 | 1024 |
|  | Relu | - | - |
|  | Linear | 1024 | 512 |
|  | BatchNorm1D | 512 | 512 |
|  | Relu | - | - |
| Projector2 | Linear | 512 | 1024 |
|  | BatchNorm1D | 1024 | 1024 |
|  | Relu | - | - |
|  | Linear | 1024 | 512 |
|  | BatchNorm1D | 512 | 512 |
|  | Relu | - | - |

The model configurations of our projectors are shown in Table 6.

## D  Limitations and Future Work.

While C-MCR offers an efficient and effective contrastive representation learning method for modalities that lack high-quality, large-scale paired data, it still necessitates an intermediate modality to associate these modalities. Exploring ways to reduce data requirements further while maintaining representation performance is an intriguing direction for future research.

## E  Social Impacts.

Although C-MCR achieves outstanding performance in audio-visual learning by connecting CLIP and CLAP, further analysis of the capability boundary of this representation is necessary before applying it to additional modalities or deploying it in practice. C-MCR only requires unpaired unimodal data during training, significantly reducing the data requirements for learning a generalizable representation. However, this also means that unsuitable and harmful data in each modality are more likely to be used for training.

