# OpenReview forum: "Connecting Multi-modal Contrastive Representations"
_NeurIPS.cc/2023/Conference — NeurIPS 2023 poster_

### Official Review · Reviewer_3nbF · 2023-06-28

**Soundness:** 3 good
**Presentation:** 3 good
**Contribution:** 2 fair
**Rating:** 5
**Confidence:** 5

**Summary:**

The SOTA approach in multi-modal learning employs contrastive loss to project embeddings from different modalities into a unified embedding space. However, models trained on different modal pairs result in disparate embedding spaces. This paper introduces a novel method to align distinct multi-modal embedding spaces by leveraging the overlapping modality they share. Additionally, it proposes several techniques to ensure a more robust alignment in the aligned embedding space. The experimental results presented in the paper demonstrate impressive performance across various tasks.

**Strengths:**

Originality: This paper introduces a novel extension to the existing CLIP approach, enhancing its capabilities by aligning distinct embedding spaces. It incorporates innovative ideas to achieve this alignment objective, contributing to the advancement of the field.
Clarity: The paper is well-structured and written in a clear style.

**Weaknesses:**

Necessity of Innovative Ideas: Although the paper introduces several innovative ideas, it would benefit from stronger evidence to support their necessity. Specifically, more clarification is needed regarding the significance of aligning the shared embedding space. What advantages does this method offer over labeling a dataset using the pretrained models utilized in this paper?
Lack of Experiment Descriptions: The experiment section would benefit from more detailed descriptions regarding the evaluation tasks and datasets used.
Additional Baselines: To establish the significance of the proposed method, it is important to include appropriate baselines trained with the same amount of data as the proposed method in the experiments.

**Questions:**

1. Could CLIP and CLAP be used to label image-audio pairs (find the image-audio pair most similar to the text embeddings, but reject the ones with distances larger than a proper threshold), using the dataset used in the proposed method, and then train an image-audio (or image-audio-text) version of CLIP? It seems that the proposed method essentially performs a similar task without explicitly creating a dataset. Inter-modality Semantic Consistency is a way to soft label noise, the step of adding Gaussian noise can be seen as a form of data augmentation, and the inter/intra alignment losses can be viewed as a type of adaptation of CLIP to three modalities.
2. Is there a specific reason for keeping the text encoder frozen during training? Would using a small learning rate for the text encoder be beneficial in any way?
3. The ablation studies indicate that softmax outperforms argmax in Consistency, but the paper doesn't explain why this improvement is related to better inter-modality semantic consistency. Is it possible that softmax serves as a form of label noise regularization (as mentioned in question 1)? Additionally, it would be helpful to include other baselines, such as the mean of the top 10 similar embeddings.
4. The baseline models were trained using different datasets compared to the proposed method, which makes it difficult to convincingly demonstrate that the proposed method outperforms the baselines. I suggest creating an audio-image dataset using pretrained CLIP and CLAP (as mentioned in question 1), and then training the baselines on this dataset. This would ensure that both the baselines and the proposed model have access to the same data.

**Limitations:**

Yes

---

> ### Author Rebuttal · Authors · 2023-08-09
>
> ## Clarification of our core innovation and the real reason our method works:
>
> The really effective part of our method is not constructing pseudo audio-image pairs using CLIP and CLAP. Instead, our methods focus on learning the alignment between text representations from CLIP and CLAP in a new space. Consequently, due to the intrinsic alignment of image-text and text-audio within CLIP and CLAP, the alignment learned from overlapping modality (e.g. text in the audio-visual setting) can be inherited by non-overlapping modality inputs (e.g. audio and visual inputs).
>
> Theoretically, the text encoders in CLIP and CLAP may introduce individual biases when encoding the same text （For example, encoding the same sentence "a dog is barking", CLIP focuses on the visual semantics of "dog", while CLAP focuses on the audio semantics of "barking".）Additionally, aggregating information from memory inevitably faces “semantic absence problem”(For example, for a text--"a mushroom", model can easily find a picture of a mushroom, but it is difficult to find audio related to“a mushroom”). It is hard to guarantee all the query data has corresponding semantics on different modalities. The accumulation of these errors diminishes the semantic coherence between the generated pseudo audio-image pairs, making it less reliable.
>
> Our ablation experiments provide strong evidence that pseudo audio-image pairs are not the key reason for our method's effectiveness. In row E of Table 4, when we removed the alignment between pseudo audio-image embeddings pairs from the inter-MCR loss, there is only a very slight performance decline (4.11, 4.13, 4.57, 4.92 → 4.09, 4.10, 4.42, 4.54). However, in row D, when we removed the alignment term between text embeddings from different MCR spaces in the inter-MCR loss, there was a significant performance drop (4.11, 4.13, 4.57, 4.92 → 3.28, 3.30, 3.69, 3.50). This verifies that the alignment learned from text-text pairs is the key to the effectiveness of our method, while the generated pseudo audio-visual pairs are merely supplementary.
>
> In the 3D-text experiments, not aligning the pseudo 3D-text pairs even results in better performance (top1, top3, top5 accuracy on Objaverse and ModelNet dataset: without aligning pseudo 3D-text: 22.6, 39.3, 47.5, 64.9, 85.0, 92.8; Full method: 19.2, 34.8, 42.9, 64.1, 83.8, 90.5). On the contrary, not aligning the representations of overlapping modality results in a severe performance decline (15.4, 27.8, 35.9, 56.1, 72.6, 79.7). These results once again confirm that the connections learned from the overlapping modality are more crucial than pseudo-data pairs.
>
> **W1 The necessity of connecting MCR:**
>
> Compared to learning or extending MCR from paired data (collected or labeled with pre-trained models), connecting MCR has the following advantages:
>
> 1. Connecting MCR enables learning multi-modal contrastive representations without large-scale and high-quality data pairs.
>
> 2. Compared with using a pre-trained model to label a paired dataset: As discussed in the “Clarification”, utilizing pre-trained models to label data suffer serve cumulative errors, resulting in less reliable semantic coherence. Our ablation study also shows the generated pseudo audio-image pair does contribute little to the final performance. Further, we conduct the experiment using "argmax" (namely label image-audio pair) to retrieve a “paired image-audio dataset” and only rely on pseudo image-audio pair to learn the alignment. The corresponding result on AVE and FlickrNet are 2.74, 2.41, 3.11, 2.97, much lower than our current methods.
>
> 3. Training efficiency and flexibility of C-MCR： More importantly, connecting MCRs only requires an extremely low training cost to learn contrastive representations between novel modalities. The low training cost also brings great flexibility to our method. Please refer to “Response to All Reviewers” for detailed discussions.
>
> **W2 descriptions regarding the evaluation tasks and datasets used:**
>
> In addition to section 4.3 of the main text, we also provide more information about the datasets used in downstream tasks and the detailed calculation process of some complex metrics in section B of the appendix. Thank you for your suggestion, we will add more details in the future version.
>
> **Q1.1 Using CLIP and CLAP to label image-audio pairs and use them to train an image-audio version of CLIP:**
>
> Please refer to "clarification" and "W1".
>
> **Q1.2 & Q2 It seems that the proposed method essentially performs a similar task without explicitly creating a dataset & Why keep the text encoder frozen?**
>
> As mentioned in "clarification", the real reason for our advanced performance is not using CLIP and CLAP to label pseudo audio-image pairs. Instead, it is learning the alignment between the text of CLIP and CLAP in a new space, and the image-audio representations can be indirectly aligned due to the intrinsic image-text and text-audio alignment in CLIP and CLAP space. Therefore, once the text encoder is learnable, the alignment of image-text and text-audio in CLIP and CLAP will be destroyed. Then, the stable alignment learned from the text cannot be maintained for image-audio input.
>
> **Q3 The softmax in consistency**
>
> Thank you for your reminder. “softmax” flexibly aggregates embeddings in memory and generate more accurate and diverse consistent representations, which indeed can be understood as a kind of label noise regularization.
>
> Although we think that the pseudo audio-image is not the key point in our method as discussed in "Clarification" and Q1, we still provide experiments that take the mean of the top 10 similar embeddings as consistent embeddings. The results are 4.08, 4.05, 4.56, 4.82, which is between the results using "softmax" and "argmax".
>
> **Q4 baseline trained on dataset labeled by CLIP and CLAP**
>
> Please refer to “Clarification” and W1.

---

> > ### Comment · Reviewer_3nbF · 2023-08-16
> >
> > - Thank you for bringing up the observations regarding rows D and E in table 4, as well as introducing the new 3D-text experiment. These findings indeed highlight that relying solely on audio-image pairs is insufficient for training a model that matches the performance of the proposed method. However, I'm curious about another method of dataset construction by labeling audio-text-image triplets using CLIP and CLAP, and training a "CLIAP" model with text as the anchor modality. Could this approach yield comparable performance to the argmax version of the proposed method?
> > - While the ablation study demonstrates that the proposed model benefits not only from audio-visual pairs but also from aligning the embedding spaces, it's important to acknowledge that a fair comparison could not be achieved when baseline methods exclude audio-visual pairs. The audio-visual information may not be enough for the proposed method, but it could be helpful for the baselines. It is quite probable that the proposed method could still outperform the baselines, but with a potentially smaller margin.
> > - The exploration of aligning the visual, sentence, and audio embedding spaces is undeniably a novel and attractive avenue. However, I feel that the evaluation limitations of the paper diminish its significance. Merely showcasing downstream audio-visual task results, which theoretically can be addressed by robust audio-visual datasets, makes the paper less useful for other researchers in understanding the alignment process. I am particularly curious about the mechanics of transforming the original embedding spaces into the new aligned space, the interplay of loss terms in regulating embedding adjustments, and whether the new embedding space captures more or less information. Without a deeper delve into the aligned embedding space, the paper's contribution to this realm seems somewhat less substantiated.

---

> > > ### Author Response · Authors · 2023-08-16
> > > **Further response to Reviewer 3nbF (1/2)**
> > >
> > > ### C1. Labeling audio-text-image triplets using CLIP and CLAP to obtain CLIAP
> > >
> > > The compared baseline AudioCLIP（for audio-visual）and ULIP 2 (for 3D-text) actually use the native or constructed audio-text-image and 3D-image-text triplet data to expand existing vision-language MCRs for more modalities. Compared to these models trained on native triplet data, our method achieves better results.
> > >
> > > Although we do not think that constructing pseudo data is the key to our method, we still construct audio-text-image triplets to extend the CLAP audio representation to CLIP space, and the retrieval results on AVE and FlickrNet are (3.04, 2.71, 3.16, 3.06), which are much lower than our method (4.11, 4.13, 4.57, 4.92) and slightly better than using "argmax" to label image-audio (2.47, 2.41, 3.14, 2.97). These results prove again that labeling pseudo data is an inefficient representation connection learning method.
> > >
> > > In addition, we discussed in our paper and rebuttal that our work is an efficient and flexible contrastive learning scheme, and the idea of constructing pseudo-datasets and using datasets to extend MCR conflicts with exploring Parameter Efficient Learning (PEL) on contrastive representation field.
> > >
> > > ### C2.1、"the proposed model benefits not only from audio-visual pairs but also from aligning the embedding spaces"
> > >
> > > This statement is inaccurate. In the 3D-text experiment, we found that aligning pseudo 3D-text data pairs brings a negative impact on learning connection. And in audio-visual, the benefit of pseudo audio-visual for learning connection is negligible. These new results in more modality settings, showcase pseudo data pairs are dispensable or even harmful to our method.
> > >
> > > ### C2.2 "It is not a fair comparison when baseline methods exclude audio-visual pairs. The audio-visual information may not be enough for the proposed method, but it could be helpful for the baselines."
> > >
> > > The mainly compared baselines, AudioCLIP and ULIP 2, use large-scale audio-visual (3D-text) data for their training. We further discuss the fairness of the comparison between our method and baseline methods as follows:
> > >
> > > **Usage of data.** All the data we use are unsupervised unimodal data. For the audio data, our data is the unimodal part of the triplets dataset used by AudioCLIP. That means, the audio data we use are just a subset of datasets used by baseline models by removing all the pair images and texts. The overall data size is approximately equal, while the data quality and annotation we use are much lower than that of baseline models.
> > >
> > > **Training costs.** The trainable parameters in our method are 60x times less than that of AudioCLIP, and since all the representations in our method can be pre-extracted and stored, the GPU memory and training time required for training are also far less than the compared baselines.
> > >
> > > **Usage of pre-trained models.** The baseline methods AudioCLIP and WAV2CLIP use CLIP, while our method uses the same CLIP and extra CLAP model.
> > >
> > > To sum up, our method can be seen as using lower-resource unsupervised data and much less training cost to achieve better contrastive learning results on different modality settings (especially for the modalities without high-quality and large-scale pairs datasets) by connecting and mining the knowledge in multiple existing representation models.
> > >
> > > Besides, we also provide the experiment of using the constructed pseudo audio-image and audio-image-text triplet to expand CLIP to ‘CLIAP’. This method of constructing pseudo datasets and then extending MCR, uses the same data, the same pre-training model, and a much more expensive training cost compared to our methods, but achieves is far inferior to our method.
> > >
> > > If you mean that the compared baselines, in addition to being trained on their original native paired datasets, also need to be further trained on the reconstructed pseudo-datasets. This far exceeds our method in terms of data, training cost, and pre-trained models. This is a unfair comparison on all aspects.

---

> > > > ### Author Response · Authors · 2023-08-16
> > > > **Further response to Reviewer 3nbF (2/2)**
> > > >
> > > > ### 3.1 "Merely showcasing downstream audio-visual task results"
> > > >
> > > > As you mentioned in your Comment 1 and we responded at “Response to all reviewers”, we add the experiment on 3D-text, which still shows the state-of-the-art performance, and we will add these results to the next version of our paper.
> > > >
> > > > ### 3.2 Downstream audio-visual tasks can be addressed by robust audio-visual datasets theoretically
> > > >
> > > > Our method is designed for modalities that are difficult to obtain large-scale and high-quality data pairs. For some modalities (e.g., audio-visual, 3D-text), it is vary expensive and difficult to obtain a large number of clear and robust data pairs.
> > > >
> > > > Experimentally, AudioCLIP uses the most advanced and largest audio-image dataset (AudioSet) for training. ULIP 2 uses image rendering and large image captioning models on the newest and largest 3D-text dataset (Objaverse) to construct a stronger 3D-image-text dataset. Although these methods have tried hard on improving the size and quality of paired datasets, they still perform inferior to our method.
> > > >
> > > > More importantly, our approach does not conflict with using large-scale data sets. Our method also has the potential to be used as initial weights or enhancement method for models trained with large-scale datasets, as we discussed in 1.4 of“Response to all reviewers”. This is also an interesting direction for further work, and we believe these can bring valuable insights into the field of contrastive learning.
> > > >
> > > > ### 3.3 “Without a deeper delve into the aligned embedding space, the paper's contribution to this realm seems somewhat less substantiated.”
> > > >
> > > > In the response to W3 of reviewer MbUd and W1 of reviewer XBwt, we analyzed the statistical metrics of the aligned representation space under different experimental settings and visualized them using T-SNE. The statistical metrics and T-SNE visualization results, with and without inter/intra-MCR, can be found in the newly submitted PDF files.
> > > >
> > > > The followings are the responses to reviewer MbUd and reviewer XBwt:
> > > >
> > > > In the original CLIP and CLAP spaces ( Figure (a) ), there exists a modality gap for both CLIP and CLAP, and the text embeddings between CLIP and CLAP are not aligned (mAP retrieval result for 1000 COCO captions is 0.14)
> > > >
> > > > In the learned CMCR space ( Figure (b) ), the modality gap between different modalities is effectively eliminated, and there is significant semantic alignment among text embeddings (mAP retrieval result is 10.41)
> > > >
> > > > In the CMCR space without the intra-MCR loss ( Figure (c) ), alignment is also observed between text embeddings (mAP retrieval result is 9.84). However, the modality gap persists, which poses challenges in maintaining the learned robust connection for image-audio inputs.
> > > >
> > > > In the CMCR space without the inter-MCR loss ( Figure (d) ), the remaining intra-MCR loss solely focuses on pulling positive instances closer, leading to the model collapse and all inputs being mapped to the same representation.
> > > >
> > > > As depicted in the visualization provided in the newly submitted PDF files, the original CLIP and CLAP space (Figure (a) ) exhibits a noticeable modality gap. After remapping by the trained projectors, the modality gap within CLIP and CLAP significantly diminishes (Figure (a) ). The L2 distance between different modalities reduces from 1.71 (CLIP), 1.99 (CLAP) to 0.53 (CLIP), 0.42 (CLAP). Closing the modality gap enables the more stable and robust connection learned from text-text to be effectively inherited by image-audio inputs.

---

> ### Author Response · Authors · 2023-08-19
> **Looking forward to further feedback**
>
> Dear Reviewer 3nbF,
>
> Thanks again for your comments. We would like to kindly remind you that we tried our best to address the concerns you raised. As the end of the author-reviewer discussion period is approaching, we would be grateful if we could hear your feedback regarding our answers to the reviews. We would be happy to discuss in detail if you have additional comments about our paper.
>
> Best regards, Authors

---

> > ### Comment · Reviewer_3nbF · 2023-08-19
> >
> > Thank you for your response. I have adjusted my rating in light of the new discussions and experiments.

---

> > > ### Author Response · Authors · 2023-08-20
> > > **Thanks for your response!**
> > >
> > > We greatly appreciate your quick reply and kind advice. We believe that your valuable comments have improved the paper, and feel free to ask more questions if you have any time. Thank you for raising the score.

---

### Official Review · Reviewer_Szdi · 2023-07-03

**Soundness:** 4 excellent
**Presentation:** 3 good
**Contribution:** 3 good
**Rating:** 7
**Confidence:** 4

**Summary:**

This paper introduced a novel technique to learn multimodal contrastive representations (MCR) without paired data from the two modalities by using a third modality to bridge between the two existing modalities. The technique assumes the existence of a learned MCR between each existing modality and the third modality, and the proposed approach brings the 2 separate MCRs together by a learned projection layer trained with both inter and intra-MCR alignments. The paper used this technique to learn an MCR between audio and images, with only unpaired audio, image and text data and existing MCR between image and text (CLIP) and between audio and text (CLAP). Then the learned MCR between audio and image was used to zero-shot on 3 audio-visual tasks: audio-image retrieval, source-localization, and counterfactual audio-image recognition. The new method is able to achieve state-of-the-art performance on all 3 tasks zero-shot, out-performing MCRs learned directly with audio-image paired data.

**Strengths:**

1.The paper studies an important and challenging problem: learning a multimodal contrastive representation between two modalities without direct paired data.

2. The proposed approach is generally quite straightforward and easy to understand. The proposed approach also only adds a very small amount of additional parameters in the projection module, and requires limited additional training.

3. The proposed approach achieved state-of-the-art performance on three different audio-visual tasks zero-shot, outperforming MCRs learned directly with paired audio-visual data.

4. The paper conducted a comprehensive ablation study that examines and justifies every design choice in the methodology.

**Weaknesses:**

Although the proposed approach does not require paired data, it does require two pre-trained that happens to bridge each of the target modalities into a third one. While this happens to be the case for audio and image modalities (where CLIP and CLAP bridges both to text), it is unclear whether this approach could be applied to a wider range of modality settings, as all experiments in this paper focused on audio-visual MCR learning.

The experiments also only included cross-modal audio-visual tasks. It would also be interesting to evaluate how good are the learned audio-visual MCR on unimodal image or audio tasks (such as image classification), since MCR like CLIP had shown great performance as unimodal image representation in addition to cross-modal performance.

**Questions:**

The descriptions in the semantic consistency ablation studies seems wrong. (Line 300-301) do you mean Row K is significantly better than Row I and J? Also, what exactly does Row J mean (with an X in consistency)?

I also have a hard time understanding the intuition behind using a softmax for semantic consistency instead of argmax (which I assume means selecting the closest example).  Why does constructing a representation of an non-existent image/audio using softmax yield better performance?

**Limitations:**

The authors did not address the limitations of the proposed method. The paper could be improved if the author discusses some limitations of their approach (such as applicability to different modality settings).

---

> ### Author Rebuttal · Authors · 2023-08-09
>
> **W1 Apply the proposed methods to other modality settings:**
>
> Please refer to “response to all reviewers” for a detailed comparison and analysis of the experiment on the 3D-text field. Briefly, our method also achieves state-of-the-art performance on the 3D-text field.
>
> **W2 Performance of the learned audio-visual MCR on unimodal tasks:**
>
> We should clarify that our method does not aim to learn stronger unimodal representation. We perform image classification on the imagenet1K dataset using the visual representation in CMCR and CLIP respectively. In our implementation, we only train a linear layer to map visual representations to classification predictions. The CLIP representations attained a top-1 accuracy of 73.1% on the ImageNet1K validation set, while our CMCR representations achieved a top-1 accuracy of 72.6%.
>
> Overall, the image classification performance of CMCR closely approximates that of the original CLIP representation. Actually, the visual representation of CMCR is just a remapping of the pre-trained CLIP representation. During the process of learning CMCR, the visual encoder would not be trained. Simple remapping of the final representation hardly enhances the ability to capture visual patterns.
>
> **Q1 Question about the Row I, J, and K in Table 4**
>
> Thank you for your reminder. We apologize for the typo in lines 300-301. The correct expression should be: "Compared to Row J, the significantly better results in Rows I 301 and K highlight......”. Row J (with an X in consistency) means randomly selecting embedding from image/audio memory to calculate intra- and inter-MCR losses, resulting in semantic-inconsistent audio and visual embeddings.
>
> **Q2 Why softmax is better than argmax:**
>
> The softmax function can be viewed as a soft information aggregation method. Using argmax to select samples involves discrete sampling within subspaces, while using softmax involves continuous sampling within subspaces. Consequently, employing softmax for interpolating the embeddings within the memory provides semantically diverse and accurate embeddings, akin to the “Mixup” technology widely used in data augmentation to enhance the diversity and accuracy of representations.

---

> > ### Comment · Reviewer_Szdi · 2023-08-15
> >
> > Thank you for your response! It's good to know that the method can also be applied to more modality combinations. I have raised my score from 6 to 7.

---

### Official Review · Reviewer_XBwt · 2023-07-05

**Soundness:** 4 excellent
**Presentation:** 3 good
**Contribution:** 3 good
**Rating:** 6
**Confidence:** 3

**Summary:**

This paper presents a Connecting Multi-modal Contrastive Representations (C-MCR) scheme that aligns the spaces of two already pre-trained MCR models by utilizing the overlapping modality between two, in a data- and training-efficient way. One interesting property of C-MCR is that aligning those two MCR spaces are feasible without the modality pairs to be aligned (e.g., image and audio) during training. To that end, two alignment objectives named Inter-MCR and Intra-MCR alignment and the Semantic Enhancement strategy are designed, while addressing the robust alignment and modality-gap issues. The Semantic Enhancement mechanism that consists of Inter-modality Semantic Consistency and Semantic Completion enables robust and comprehensive alignment. In addition, the two proposed objectives are the key to the semantic-enhanced inter- and intra-MCR connection in the C-MCR framework. As an instantiation of the C-MCR framework, the audio-visual-aligned contrastive representation is obtained using CLIP and CLAP models, and its effectiveness is verified by a series of experiments on various audio-visual tasks in a zero-shot manner.


**Strengths:**

- The motivation for the C-MCR is clear and strong. Achieving the aligned MCR spaces without using the explicit modality pairs which can be expensive in some situations seems to be a clear advantage.

- The idea for aligning the MCR spaces including semantic completion mechanism and MCR alignment objectives is simple and straightforward, while it seems to be effective even lack of direct audio-visual pairs during training.

- The presentation of the work is quite clear and the paper is generally well written. The codes are included in the submission.

**Weaknesses:**

- It would be desirable to make analysis on whether the two projectors trained to indirectly align audio-visual modalities via the proposed C-MCR scheme can also improve the alignment between image and language / audio and language compared to simple CLIP or CLAP. If it could, such discovery can strengthen the contribution of this work.

- In this work, connecting only two MCR representations is explored. Discussion on whether the proposed scheme is seamlessly applied to aligning three or more MCR modalities would be helpful.

- Sensitivity analysis on some important hyper-parameters (e.g., temperature scaler $\tau$ and loss weight $\lambda$) is missing. How those sets of hyper-parameters are determined?

- For the projection layers, two layer MLPs with activations are adopted. Is the alignment ability affected by the depth of the projection layer?  Would like to see the performance changes when using single-layer MLP or MLPs with more than two.

**Questions:**

- The equations for the contrastive loss functions and the Softmax of similarity terms are too redundant. The only variations are the input and temperature parameters. To improve clarity, it is suggested to define the softmax of similarity terms and the form of contrastive losses as new functions and clean the following equations.

- If time permits, I hope to see whether the proposed C-MCR method can be generalized to align the MCR representations between 3d point clouds and language via 2d image modality (as explained in the introduction section).

**Limitations:**

Limitations and potential negative societal impact are addressed.

---

> ### Author Rebuttal · Authors · 2023-08-09
>
> **W1 Whether trained projectors can improve the alignment between image and language/audio and language:**
>
> As depicted in the visualization provided in the newly submitted PDF files, the original CLIP and CLAP space (Figure (a) ) exhibits a noticeable modality gap. After remapping by the trained projectors, the modality gap within CLIP and CLAP significantly diminishes (Figure (a) ). The L2 distance between different modalities reduces from 1.71 (CLIP), 1.99 (CLAP) to 0.53 (CLIP), 0.42 (CLAP). Closing the modality gap enables the more stable and robust connection learned from text-text to be effectively inherited by image-audio inputs.
>
> **W2 & Q2 Discussion on employing the proposed scheme to 3D-Language or more modalities**
>
> Please refer to “response to all reviewers” for a detailed comparison and analysis of the experiment on the 3D-language field. Our method also achieves advanced performance in this field.
>
> As for applying our method to three or more MCR modalities, due to time constraints, we are unable to conduct this evaluation experiment. Such an evaluation would require collecting multiple pre-trained models for different modalities, datasets for each modality, as well as appropriate validation tasks, validation datasets, and comparable baselines for evaluating different modality combinations. However, in theory, connecting multiple modalities only requires constructing semantically enhanced embeddings through each overlapping modality anchor and simultaneously optimizing inter-MCR losses for each overlapping modality and intra-MCR losses within each MCR. Several studies [1][2][3] have demonstrated that simultaneous optimization of multiple contrastive learning losses can lead to a unified representation space. Given the successes of C-MCR in audio-visual and text-3D representations, as well as the successful examples of simultaneously optimizing multiple contrastive learning losses, our approach is promising for generalizing to more modalities.
>
> [1] ULIP-2: Towards Scalable Multimodal Pre-training For 3D Understanding. arxiv 2023
>
> [2] Any-to-Any Generation via Composable Diffusion. arxiv 2023
>
> [3] Imagebind: One embedding space to bind them all. CVPR 2023
>
> **W3 Sensitivity analysis and selection of hyper-parameters.**
>
> The temperature scaler is set to 1/100 which is the final learned temperature scaler in CLIP. The loss weight of 0.1 is set to ensure that the absolute values of the two losses remain comparable. Here are some sensitivity analyses of these two hyper-parameters.
>
> |  Temperature | AVE   |   Flickr  |
> | :------------: | :------------: | :------------: |
> |  1/50  |   4.11  |    4.64  |
> |  1/75  |   4.18   |   4.84  |
> |   1/100 |   4.11 |     4.57   |
> |  1/125  |  4.11  |    4.53   |
> |  1/150  |  4.08     | 4.56 |
>
> |  Loss factor  | AVE  | Flickr  |
> | :------------: | :------------: | :------------: |
> |  0.05    |  4.07    |  4.68  |
> |  0.10   |   4.11   |   4.57   |
> |  0.20   |   4.08    |  4.68  |
> |  0.30 |     4.17   |   4.96  |
> |  0.40  |    4.09   |   4.85  |
>
> Overall, our method is not sensitive to temperature scaler and loss weight. By carefully tuning these two hyper-parameters, the retrieve performance on AVE and Flickr-SoundNet even further improves from 4.11, 4.57 to 4.17, 4.96.
>
> **W4 Effect about the number of layers in MLP:**
>
> Here is the ablation study about the layer number of the MLP.
>
> |  Layer  | AVE  | Flickr  |
> | :------------: | :------------: | :------------: |
> |1-layer | 3.84  |    4.47 |
> |2-layer  |4.11   |   4.57 |
> |3-layer | 4.20  |   4.77 |
> |4-layer  |4.28  |   4.74 |
> |5-layer | 4.26   |   4.60 |
>
> Overall, our scheme is also not sensitive to the layer number of the MLP. By carefully tuning the number of linear layers in the MLP, the retrieve performance on AVE and Flickr-SoundNet further improve from 4.11, 4.57 to 4.28, 4.74.
>
> **Q1 Too redundant equation:**
>
> Thanks for your constructive advice, we will revise it in the future version.

---

> > ### Comment · Reviewer_XBwt · 2023-08-16
> > **Post-rebuttal comment**
> >
> > I acknowledge the authors' effort in rebuttal. I have gone through the comments from other reviewers and the rebuttal. The newly introduced 3d-language experiments can make the proposed framework more persuasive and general. In addition, thank you for presenting additional ablation study and analysis in rebuttal to better understand the method. I think that most of my concerns have been properly addressed, so I would like to keep my favorable rating.
> >
> > It would be more valuable to include discussion on connecting three or more modalities presented in the rebuttal for the final revision, and to resolve the fairness issues in experiments as raised by other reviewer.

---

### Official Review · Reviewer_5S9J · 2023-07-06

**Soundness:** 3 good
**Presentation:** 3 good
**Contribution:** 2 fair
**Rating:** 6
**Confidence:** 4

**Summary:**

- This paper proposes C-MCR, a multimodal contrastive framework that aligns multimodal representations of overlapping modality pairs to learn effective representations of the non-overlapping modality pairs. Specifically, the authors concentrate on learning the inherent semantics of visual-audio data solely through language-visual and language-audio pairs. The authors developed lightweight projectors, semantic enhancement techniques, and inter-intra-modality alignment objectives. Experiments are well designed on various downstream tasks in a zero-shot manner for C-MCR with different metrics, and C-MCR demonstrates superior performance compared to the chosen baselines. The authors further conduct ablation studies on the proposed components to evaluate the effectiveness of C-MCR.

---------------------------

My score is updated after reading the authors's rebuttal and the reviews from my peer reviewers.

**Strengths:**

- The motivation to connect modality representations and reduce reliance on high-quality modality pairs is strong, and the proposed method is straightforward and clearly described.
- The authors have presented illustrative figures to explain their framework
- The proposed method demonstrates notable performance improvements over other baselines.
- Ablation studies are well-defined and provide clear insights into the effectiveness of the proposed modules

**Weaknesses:**

- Figure 1 is highly informative, but it may cause confusion when simply referred to as “As illustrated in Figure 1”. Adding more detailed text or subsection numbers to label the components would enhance the clarity for the readers.
- It would be more promising if the approach section could be generalized for additional modalities to demonstrate the extensibility and significance of the proposed methods in various applications involving multimodal data.
- Related works could also include additional multimodal contrastive learning frameworks [a1, a2]
- The proposed modality pair alignment is very interesting, however, the idea of projectors and contrasting components has already been explored in the past.

[a1] "Contrastive multiview coding." Computer Vision–ECCV 2020

[a2] "Geometric multimodal contrastive representation learning." International Conference on Machine Learning. PMLR, 2022.”

**Questions:**

- The current method is focusing on visual audio with text. Is it possible to generalize it with different or more modalities? In this case, there could be multiple overlapping pairs, rather than just two modality pair with one modality as the overlapped anchor.
- Section 3.4 “there are (x_i - y_i)^T……”. The wording could be clarified more
- Section 4.3 Audio-Visual Source Localization “score is token as the final prediction”, the word token causes some confusion

**Limitations:**

- The authors could provide a more comprehensive discussion on the limitations of the proposed method and future works. For example, they could explore the possibilities of extending the method to accommodate additional modalities and the application in other domains.

---

> ### Author Rebuttal · Authors · 2023-08-09
>
> **W1: Too informative Figure 1**
>
> Thank you for your constructive suggestion. We will add subsection numbers to the subfigures in Figure 1 and make the necessary adjustments to the reference statements in the main text accordingly.
>
> **W2 & Q1: Generalize our approach to different and more modalities.**
>
> For generalization to different modalities, please refer to “response to all reviewers” for the experiment that employs our methods in the 3D-text field. Briefly, our method also shows advanced performance in these modalities.
>
> As for applying our method to three or more MCR modalities, due to time constraints, we regret that we were unable to conduct an evaluation experiment on applying our method to three or more MCR modalities. Such an evaluation experiment requires collecting multiple pre-trained models for different modalities, datasets for each modality, appropriate validation tasks, validation datasets, and comparable baselines for evaluating different modality combinations. However, in theory, connecting multiple modalities only requires constructing semantically enhanced embeddings through each overlapping modality anchor and simultaneously optimizing inter-MCR losses for each overlapping modality and intra-MCR losses within each MCR. Several studies [1][2][3] have demonstrated that simultaneous optimization of multiple contrastive learning losses can lead to a unified representation space. Given the successes of C-MCR in audio-visual and text-3D fields, as well as the successful examples of simultaneously optimizing multiple contrastive learning losses, our approach is promising for generalizing to more modalities.
>
> [1] ULIP-2: Towards Scalable Multimodal Pre-training For 3D Understanding. arxiv 2023
>
> [2] Any-to-Any Generation via Composable Diffusion. arxiv 2023
>
> [3] Imagebind: One embedding space to bind them all. CVPR 2023
>
> **W3: Related works:**
>
> Thank you for your valuable suggestion. We will add these references to our related works.
>
> **W4: the idea of projectors and contrasting components:**
>
> To the best of our knowledge, we are the first to propose learning contrastive representation by connecting existing contrastive representations, rather than training on large-scale data pairs. Moreover, the role of a projector in our method is a highly parameter-efficient structure for mapping representations. Our method is a very parameter-efficient learning method that requires extremely low training cost, which is also first studied in multi-modal contrastive representation learning. Please refer to the "Response to all reviewers" section for more analysis of the training efficiency and flexibility of our method.
>
> Could you kindly provide some references about “the idea of projectors and contrasting components” for further discussion?
>
> **Q2 The wording in Section 3.4 could be clarified more:**
>
> Thanks for your advice, we will revise this in a further version.
>
> **Q3 word “token” may cause confusion:**
>
> Thanks for your reminder, We will modify it to“score is selected as the final prediction”
>
> **Limitations and Future Work:**
>
> We will add a discussion about connecting more than three modalities simultaneously and the potential of assembling knowledge from different MCRs in limitations and future Work in future versions.

---

> > ### Comment · Reviewer_5S9J · 2023-08-14
> > **Replies after reading the authors' responses**
> >
> > Thanks to the authors for the thorough responses. Since new results on 3D-text have been added, I would like to raise my score to 6. However, after reading the discussions between authors and other reviewers, I am not fully convinced of the presented evaluations' fairness and the proposed approach's applicability to more than 2 modalities, especially when no direct connections to a common modality (like text) are available.

---

### Official Review · Reviewer_MbUd · 2023-07-07

**Soundness:** 3 good
**Presentation:** 3 good
**Contribution:** 2 fair
**Rating:** 5
**Confidence:** 4

**Summary:**

This paper focuses on extending existing Multimodal Contrastive Representation (MCR) to more modalities without massive high-quality data pairs. To this end, they propose a training-efficient method for learning a new MCR without paired data. Specifical, they connect one existing MCR pre-trained on modality $(A, B)$ and one existing MCR pre-trained on modality $(B, C)$, to obtain the new MCR pre-trained on modality $(A, C)$. As a result, they connect the pre-trained CLIP and CLAD models to derive audio-visual data, which achieves significant retrieval performances on AVE and Flickr-SoundNet.

**Strengths:**

1.	**The writing is clear and well-motivated.** It is necessary to extend multimodal contrastive representation to more modalities without massive high-quality data pairs.

2.	**Technically sound and easy to understand.** The semantic-enhanced inter- and intra-MCR connection method is flexible and training-efficient.

3.    **The results are impressive.** Without any paired audio-visual data, they achieve state-of-the-art performance on six datasets. More specifically, Tab.2 and Tab.3 show that C-MCR surpasses existing methods by a large margin.

**Weaknesses:**

1.  **The main results are not sufficient**.  The current experimental results are only experiments on the visual-audio task. This is not enough to support the claim of connecting multi-modal contrastive representation.  This article should provide more modality alignment experiments, e.g. (3D point, image)->(text, image), (text, image)->(image, sketch/RGB-D).

2.  **The comparison experiments are not fair**. As shown in Table 2 and Table 3, a column should be added to the table to discuss which pre-trained models are used. It is unfair to only compare performance with different pre-trained model usage settings.

3.   **Lack of thorough analysis of the proposed method**. The inter-MCR and intra-MCR are needed for visualizing by T-SNE to understand the mechanism of modules. In addition, the intra-variance and inter-variance between different modalities should be measured w or w/o inter/intra-MCR.

4.  **The proposed method is trivial**. To some extent, there have existed similar methods [1][2][3][4] that align the semantic-enhanced embeddings across different modality spaces. Can you provide a more thorough comparison to illustrate the novelty of your work?


[1] Text-only Training for Image Captioning using Noise-injected CLIP. EMNLP 2023.

[2] I Can't Believe There's No Images! Learning Visual Tasks Using Only Language Data. Arxiv 2022.

[3] From Association to Generation: Text-only Captioning by Unsupervised Cross-modal Mapping. IJCAI 2023.

[4] DeCap: Decoding CLIP Latents for Zero-Shot Captioning via Text-Only Training. ICLR 2023.


**Questions:**

As shown in Weaknesses.

---

> ### Author Rebuttal · Authors · 2023-08-09
>
> **W1: The main results are not sufficient:**
>
> Please refer to “response to all reviewers” for the experiment of learning 3D-text representation via image-text and image-3D pre-trained model.
>
> **W2: Unfair comparison experiments:**
>
> Thank you for your suggestion. We will add a column in Table 2 and 3 to discuss the used pre-trained method.
>
> But we should clarify that the results presented in Table 2 and 3 primarily aim to evaluate the alignment between fine-grained visual objects and audio of audio-visual representations. The comparison with previous sound source localization or counterfactual audio-image recognition methods only serves as a baseline reference, the focus is the comparison between other audio-visual representations such as AudioCLIP and WAV2CLIP.
>
> We employed these three audio-visual representations (AudioCLIP, WAV2CLIP, and our C-MCR) for zero-shot audio-visual source localization and counterfactual audio-image recognition tasks via the same object detector and pipeline. Under identical settings, our audio-visual representations outperform AudioCLIP and WAV2CLIP on all four datasets across two tasks. Additionally, it is worth noting that AudioCLIP and WAV2CLIP exhibit significant performance fluctuations across different datasets, indicating a lack of generalization. In contrast, our method consistently demonstrates advanced performance across all four datasets.
>
> **W3: Lack of thorough analysis of the proposed method:**
>
> Some statistical metrics, as well as T-SNE visualization results, with and without inter/intra-MCR, are provided in the newly submitted PDF files.
>
> In the original CLIP and CLAP spaces ( Figure (a) ), there exists a modality gap for both CLIP and CLAP, and the text embeddings between CLIP and CLAP are not aligned (mAP retrieval result for 1000 COCO captions is 0.14)
>
> In the learned CMCR space ( Figure (b) ), the modality gap between different modalities is effectively eliminated, and there is significant semantic alignment among text embeddings (mAP retrieval result is 10.41)
>
> In the CMCR space without the intra-MCR loss ( Figure (c) ), alignment is also observed between text embeddings (mAP retrieval result is 9.84). However, the modality gap persists, which poses challenges in maintaining the learned robust connection for image-audio inputs.
>
> In the CMCR space without the inter-MCR loss ( Figure (d) ), the remaining intra-MCR loss solely focuses on pulling positive instances closer, leading to the model collapse and all inputs being mapped to the same representation.
>
> **W4: The proposed method is trivial.**
>
> Our approach differs significantly from these methods in terms of task definition, overall methods, and specific approaches for addressing the modality gap.
>
> In terms of task definition, the mentioned papers focus on learning text generation downstream tasks using text inputs and evaluating with visual input, while our approach aims to construct a contrastive representation for novel modalities by connecting two existing MCR spaces. Our method focuses on the representation learning problem rather than a specific downstream task. This fundamental difference sets our problem setting apart from prior works. Additionally, the new experiments involving 3D-text further highlight that our approach fundamentally revolves around representation learning rather than specific downstream tasks.
>
> Regarding overall methods, although the mentioned works also discuss the modality gap in contrastive representations, their methods only aim to address the gap in a single MCR space. In contrast, our approach emphasizes building a connection between two MCR spaces while considering the modalities gaps within both spaces. Our method proposes an effective semantic enhancement approach and facilitates cross-space and cross-modal alignment, distinguishing our work from these papers.
>
> Even only considering the specific methods for addressing the modality gap, our proposed solution stands out from previous approaches. In [1][2], random noise is added to text representations, aiming to eliminate the modality gap through the non-directional offset introduced by these noises. But the offset between the modality gap is directional, the non-directional offset is unreliable for the modality gap. On the other hand, [3][4] maintain a set of text embeddings during testing, retrieving semantically related embeddings from a text bank as input of downstream head, which inevitably incurs additional inference costs. In contrast, We design an intra-MCR alignment loss by analyzing the main reason for the modality gap (e.g. the “pulling apart” term in the contrastive learning loss). Our method can directionally close the modality gap without introducing extra inference costs. Our unique approach to the modality gap problem differs from these works in terms of motivations, detailed algorithms, and advantages.

---

> > ### Comment · Reviewer_MbUd · 2023-08-16
> > **Post-rebuttal after reading the response.**
> >
> > Thanks for your response! The experiments on the 3D-text representation are convincing. And the visualization (T-SNE) of inter/intra-MCR provides more insight for the community. Thus, I would like to raise my score to 5 (borderline accept).

---

### Author Rebuttal · Authors · 2023-08-09

# Response to all reviewers

**1 Experiment of 3D-text**

1.1 Performance comparison:

By connecting text-image representations of CLIP and image-3D representation of ULIP 2 (Objaverse), we successfully learned a contrastive 3D-text representation without explicitly using any 3D-text data. We evaluated the performance of our 3D-text representation on two  3D object classification datasets: ModelNet40 and Objaverse-LVIS. Here is the comparison of our CMCR (3D-text) with previous 3D-text representations:

Results on Objaverse-LVIS:

|   |   |Top1 |    Top3 |    Top5 |
| :------------: | :------------: | :------------: | :------------: | :------------: |
| ReCon [1] |  ICML2023  |   1.1 | 2.7 | 3.7 |
| CG3D [2] |  arxiv2023  |   5.0 | 9.5 | 11.6 |
|  ULIP [3] |   CVPR2023  |  6.2 | 13.6 |17.9  |
|  ULIP2 (objaverse) [4] | arxiv2023  |  6.3 | 11.9 | 34.5 |
|  C-MCR (3D-text) | - |  **22.6** | **39.3** | **47.5** |

Results on ModelNet40:

|   |   |Top1 |    Top3 |    Top5 |
| :------------: | :------------: | :------------: | :------------: | :------------: |
| ReCon [1] |  ICML2023  |   61.2 | 73.9 | 78.1 |
| CG3D [2] |  arxiv2023  |   48.7 | 60.7 | 66.5 |
|  ULIP [3] |   CVPR2023  | 60.4 | 79 | 84.4  |
|  ULIP2 (objaverse) [4] | arxiv2023  |  **74.0** |  86.5 | 90 |
|  C-MCR (3D-text) | - |  64.9 | **87.0** | **92.8** |

Overall, our C-MCR (3D-text) achieves advanced performance on ModelNet40 in a zero-shot setting and demonstrates significantly better performance on the more challenging and long-tail Objaverse-LVIS dataset.

The most important comparison in this table is the comparison between our C-MCR (3D-text) and ULIP 2 (Objaverse). ULIP 2 constructs image-3D-text triplets on the Objaverse dataset to learn image-text-3D representations, where the learned text-3D representation is used for classification. Our C-MCR only utilized the image and 3D encoders from ULIP 2, and by connecting them with CLIP, we obtained a new 3D-text representation. Remarkably, without using any 3D-text data, our C-MCR outperformed ULIP2, particularly on the more challenging and long-tailed Objaverse-LVIS dataset, which demands higher generalization performance.

[1] Contrast with reconstruct: Contrastive 3d representation learning guided by generative pretraining. ICML 2023

[2] Clip goes 3d: Leveraging prompt tuning for language grounded 3d recognition. arxiv 2023

[3] Ulip: Learning unified representation of language, image and point cloud for 3d understanding. CVPR 2023

[4] ULIP-2: Towards Scalable Multimodal Pre-training For 3D Understanding. arxiv 2023


1.2 Implementation Details:

We employ the 1.2M training images in ImageNet1K for training. The 3D object point clouds from the training set of Objaverse are utilized to construct memory (0.8M samples in total). For the text memory, we use the same dataset combination (2.3M) described in Section 4.1 of our main text.

1.3 Pre-training weight comparison：

ULIP 2 uses the pre-trained BLIP-2 to label text for image-3D pairs, its visual and text encoder utilize an advanced version of CLIP, namely SLIP. ULIP 2 (Objaverse) is pre-trained on Objaverse 3D-image-text triplets.

Our C-MCR uses the pre-trained CLIP ViT-B/32 model, and the pre-trained visual and 3D encoders of ULIP 2 (Objaverse).

1.4 More discussion about the comparison between ULIP 2 and our C-MCR

Our C-MCR builds upon ULIP 2, even without using the pre-trained text encoder of ULIP 2，
there may be a concern that the 3D encoder might implicitly learn the knowledge in 3D-text data. From this perspective, our approach can be understood as learning stronger text representations by connecting CLIP to ULIP 2 (image-3D only), and using the newly obtained text representation to replace the original one derived from paired data. By connecting with CLIP via our method, the alignment of 3D-text in ULIP 2 is further boosted. This is a very intriguing and insightful discovery, highlighting the potential of our method to effectively reorganize knowledge of one contrastive representation and enhance its internal alignment under the guidance of another contrastive representation. Our method not only can connect two MCRs to construct contrastive representations between novel modalities, but also can assemble knowledge from different MCRs to enhance alignment in MCR which unified align three and more modalities.

**2 More discussion about the training efficiency and flexibility of C-MCR**

Connecting MCR requires an extremely low training cost. Contrastive learning is known for its huge training cost due to the large batch size required, while our method only needs to train two simple projector layers, and all the representations used can be pre-extracted and saved, thus achieving extremely low training costs. Taking learning 3D-text contrastive representations as an example, our method requires only 4G GPU memory and takes only 40 minutes for training on a single 3090 GPU. We believe that this paradigm of mining new knowledge from existing MCRs with minimal training cost can provide valuable inspiration for Parameter-Efficient Learning (PEL) in representation learning.

The extremely low training cost brings great flexibility. Any newly released stronger multi-modal contrastive pre-trained model and high-quality large-scale datasets can be easily used by our method without worrying about the training cost increase brought by the data and model scale. Combining this flexibility with the potential for creating new contrastive representations and assembling multiple MCRs (discussed in "the comparison between ULIP 2 and our C-MCR"), our approach holds significant practical value in integrating knowledge from various advanced MCRs.

---

### Author Response · Authors · 2023-08-16
**Thanks to all reviewers**

We would like to thank all reviewers for their constructive comments and quick responses. We believe that your valuable comments have improved the paper,  and feel free to ask more questions if you have any time. Thanks again for the efforts of all reviewers.

---

### Decision · Program_Chairs · 2023-09-21

**Decision:**

Accept (poster)

**Comment:**

This paper brought an active discussion. The latest results shared by the authors on the task of 3D-text representation helped emphasizing the novelty and performance of the proposed approach. The paper presents a novel approach to integrate representations from two bimodal models to allow the connection of two non-overlapping connections. The paper includes extensive set of experiments, visualizations analyses. The discussion allowed to clarify reviewer's concerns, with an agreement among reviewers that this paper is worth publication at Neurips. I support this assessment.